# PMixUp: Simultaneous Utilization of Part-of-Speech Replacement and Feature Space Interpolation for Text Data Augmentation

## Abstract

Data augmentation has become a de facto technique in various NLP tasks to overcome the lack of a large-scale, qualified training set. The previous studies presented several data augmentation methods, such as replacing tokens with synonyms or interpolating feature space of given text input. While they are known to be convenient and promising, several limits exist. First, prior studies simply treated topic classification and sentiment analysis under the same category of text classification while we presume they have distinct characteristics. Second, previously-proposed replacement-based methods bear several improvement avenues as they utilize heuristics or statistical approaches for choosing synonyms. Lastly, while the feature space interpolation method achieved current state-of-the-art, prior studies have not comprehensively utilized it with replacement-based methods. To mitigate these drawbacks, we first analyzed which POS tags are important in each text classification task, and resulted that nouns are essential to topic classification, while sentiment analysis regards verbs and adjectives as important POS information. Contrary to the aforementioned analysis, we discover that augmenting verbs and adjective tokens commonly improves text classification performance regardless of its type. Lastly, we propose PMixUp, a novel data augmentation strategy that simultaneously utilizes replacement-based and feature space interpolation methods. We examine that they are new state-of-the-art in nine public benchmark settings, especially under the few training samples.

## 1 Introduction

**Background and Motivation** Recent improvements in deep neural networks have empowered remarkable advancements in various Natural Langauge Processing (NLP) tasks such as text classification (Minaee et al., 2021), question answering (Rogers et al., 2021), and natural language inference (Bowman & Zhu, 2019). However, these supreme performances rely on large, qualified training sets under the supervised regime. Under the circumstance where the machine learning practitioners have a limited number of training samples, the models tend to suffer from overfitting and fail to function with the expected performance. As acquiring large-scale, qualified training samples requires a particular amount of resource consumption, there have been numerous studies to escalate the model's performance under the limited number of training samples in NLP tasks (Hedderich et al., 2020). One promising approach is data augmentation, which generates new training samples by modifying original training samples through transformations (Chen et al., 2021). An underlying motivation of data augmentation is to augment original training samples while the transformed text data sustain the original sample's overall semantics (Bayer et al., 2021).

Several studies on data augmentation techniques proposed back-translation (Sennrich et al., 2015) and word replacements with predictive language models (Anaby-Tavor et al., 2020; Yang et al., 2020b). As these approaches require the massive cost of implementation (i.e., a well-trained translation model for back-translation), academia sought more lightweight augmentation methods. Several studies presented replacement-based augmentation methods where tokens are replaced with synonyms fetched from large-scale dictionaries (i.e., WordNet) as the synonyms are less likely to affect the semantics of original training samples (Bayer et al., 2021; Feng et al., 2020; Wei & Zou, 2019). While replacement-based augmentation methods escalate the model's performance, it risks creating

unrealistic or non-conforming samples. This drawback motivated the development of feature space interpolation methods that transform a given input in a feature space rather than directly changing tokens (Sun et al., 2020; Chen et al., 2020b; Shen et al., 2020). It has become state-of-the-art in data augmentations on text classification.

**Main Idea and Its Novelty** When we scrutinize replacement-based approaches, we figure out that most of them did not precisely consider Part-of-Speech (POS) information. We presume there exists a particular POS that highly correlates to the text classification performance, but it was not actively considered in past studies. Furthermore, we hypothesize that essential POS information would differ along with text classification types (topic classification and sentiment analysis) presented in (Sun et al., 2019). As these tasks have varying characteristics among themselves, we expect the augmentation strategies should be considered different respective to each task. But, it was not considered in the prior works. Lastly, replacement-based methods and feature space interpolation approaches have not been simultaneously considered. The replacement-based methods have an advantage in providing more contextual augmentation results as they directly transform the word token with its synonyms. On the other hand, feature space interpolation methods are also effective because they create an infinite amount of new augmented data samples as they do not transform the token, but instead, add perturbations to the feature vector. Then, the following question becomes our primary motivational question: *What if we simultaneously utilize both replacement-based and feature space interpolation methods?*

To this end, we propose a novel data augmentation method denoted as Part-of-speech MixUp (**PMixUp**), which replaces tokens belonging to the particular POS tags and applies feature space interpolation in sequential order. We presume simultaneous utilization of both replacement-based and feature space interpolation methods would further escalate the classification performance. As a preliminary analysis, we firstly examine which POS information influences two text classification types the most and validate whether augmenting tokens that do not belong to important POS tags helps maintain the original sentence's overall semantics. We then examined its effectiveness in nine benchmark datasets under the various training sample sizes per class and resulted in the proposed PMixUp accomplishing new state-of-the-art in given public benchmark settings. We hereby highlight our work's novelty in the following aspects. First, our work firstly analyzes how POS information becomes important following each classification type. Second, to the best of our knowledge, our study is the first attempt to simultaneously utilize replacement-based and feature space interpolation methods for text augmentation.

**Key Contributions**

- We empirically discovered that nouns serve as critical label determinants in topic classification while verbs and adjectives play essential roles in sentiment analysis.

- We figured out that replacing tokens that do not belong to the aforementioned important POS tags at both text classification tasks contributes to maintaining the original sentence's meaning and elevating the test classification performance. Instead, we discovered a common trend across the two tasks that replacing tokens under verbs and adjectives POS tag escalated the test performance at most, and we analyze the reason for this phenomenon as the low interference with core semantics of original text data when replaced.

- We present PMixUp, a novel data augmentation technique that combines POS-guided replacement and feature space interpolation methods. Upon nine public benchmark datasets, we examined the proposed PMixUp's supreme performance of classification performance escalation rather than prior works; thus, it becomes a new state-of-the-art in given settings.

- We discovered that PMixUp is especially effective under a few training samples per class. We further observed that every data augmentation method's beneficial impact decreases when there are many training samples.

- Lastly, we scrutinized the supremacy of PMixUp derives from larger knowledge capacity, that our method makes the model acquire more fruitful understanding at given data.

## 2 RELATED WORKS

**Replacement-based Data Augmentation** The replacement-based augmentation strategies generate text samples by replacing particular tokens or words with synonyms (Wei & Zou, 2019). This

augmentation technique presumes that the overall semantic information of a sentence remains unchanged as long as particular words are replaced with their synonyms. Synonyms of the tokens to replace are typically matched from a pre-defined source such as WordNet (Miller, 1995), Thesaurus (Jungiewicz & Smywiński-Pohl, 2019), or from a pre-trained Language Model such as BERT (Devlin et al., 2018). (Kolomiyets et al., 2011) presented a headword replacement technique, which creates augmented training samples by substituting temporal expression words with synonyms from WordNet and Latent Words Language Model. (Wei & Zou, 2019) proposed a random replacement technique that randomly chooses $N$ words from a given sentence and substitutes them with randomly chosen synonyms. (Feng et al., 2020) suggested GenAug technique which utilizes RAKE score (Rose et al., 2010) for keyword extraction and substitutes these keywords with synonyms at WordNet. Recently, several studies proposed POS-based replacement techniques. These studies chose particular POS (i.e., nouns, verbs) as a replacement target because they have a higher probability of having synonyms than the others (Marivate & Sefara, 2020; Jungiewicz & Smywiński-Pohl, 2019). While these approaches share a similar motivation with our study of replacing particular POS, our study differs from them because we postulate having a higher probability of having synonyms does not simply implicit effective augmentation performance. Our study hypothesizes there certainly exists a particular POS along with text classification's types, and considering this understanding correlates to precise augmentation performance.

**Feature Space Interpolation** Feature space interpolation techniques transform a given text sample's feature representation to generate augmented training samples. An underlying motivations of feature space interpolation techniques are follows: producing synthetic samples from the input level poses the risk of dealing with unrealistic samples that leads to a performance drop; thus, there should be an alternative augmentation method that does not directly transform original sample (DeVries & Taylor, 2017). Upon this motivation, (Ozair & Bengio, 2014; Bengio et al., 2013) showed that augmenting in feature space leads to a higher chance of producing realistic examples compared to that in data space or input level. Such augmentation strategies include adding noise, erasing, or interpolating instances of input text. (Kurata et al., 2016) proposed an approach that augments text by randomly perturbing the encoded vectors of input. (Shen et al., 2020) suggested CutOff, which erases part of the feature vector to produce numerous new samples with noise. Interpolation combines two different training samples to create a new one. (Kumar et al., 2019) tried joining the first $[CLS]$ tokens of two training examples to form a new instance. (Sun et al., 2020) introduced Mixup Transformer, which interpolates the vectors obtained from the last layer of the transformer architecture. Similarly, (Chen et al., 2020b) introduced TMix, which interpolates at specific layers within a transformer architecture, and showed that combining hidden vectors at the 7th, 9th, and 12th layers containing the syntactic and semantic information improved the preceding interpolating methods. With respect to the motivation of feature space interpolation techniques, we presumed utilizing these methods as well as replacement-based augmentation techniques might further escalate the classification performance.

## 3 PRELIMINARIES

**Text Classification and Datasets** We categorize the datasets into two types: topic classification and sentiment analysis, following the taxonomy and definition proposed by (Sun et al., 2019). For topic classification, we utilized the datasets StackOverFlow, BANKING, DBPedia, r8, AGNews and Ohsumed, while using IMDB, Amazon Polarity Reviews and Yelp Reviews for sentiment analysis. The detailed summaries are as below:

- **StackOverFlow** : Dataset published in Kaggle.com. We follow the works of (Xu et al., 2015) in using a processed version with 20 classes and 1,000 samples for each class.

- **BANKING** : A dataset under BANKING domain, consisting of 13,083 customer service queries with 77 intent classes (Casanueva et al., 2020).

- **DBPedia** : An ontology classification dataset (Zhang et al., 2015)

- **r8** : Dataset consisting of news documents from 8 most popular classes of Reuters-21578 corpus (Debole & Sebastiani, 2005)

- **AGNews** : A large-scale collection of news articles with 4 classes (Zhang et al., 2015).

- **Ohsumed** : A subset of the MEDLINE database. The dataset is composed of 23 Medical Subject Headings classes of cardiovascular diseases group, and we use the commonly used subset from (Yao et al., 2017).

- **IMDB** : A movie review dataset of 50k full length reviews (Maas et al., 2011).

- **Amazon Polarity Reviews** : Dataset consisting of reviews from Amazon, with binary polarity labels. (Zhang et al., 2015)

- **Yelp Reviews** : The polarity-labeled version of dataset obtained from Yelp Dataset Challenge in 2015. (Zhang et al., 2015)

**Classifier and Evaluation** We utilize pre-trained BERT (Devlin et al., 2018) as a backbone language model, and we used implementations provided in Huggingface (Wolf et al., 2019). For training the classifier, we set the learning objective with cross-entropy loss. We trained the model for 15 epochs with AdamW optimizer (Loshchilov & Hutter, 2017) under the learning rate of 4e-5. Following the evaluation method utilized in various studies regarding text classification and data augmentations (Fujino et al., 2008; Mishu & Rafiuddin, 2016; Ollagnier & Williams, 2020), we employed the F1 score at the test set as an evaluation metric. For the reproducibility of our study, we publicize the code implementations in the supplementary materials.

## 4    DISCOVERING KEY FACTORS OF TEXT CLASSIFICATION

**Setup** As a first and foremost analysis, we investigate key factors that contribute to classifying the intent or label of a given text at each text classification type. We expect an effective data augmentation technique shall sustain a given text's general semantics after replacing particular tokens (Chen et al., 2021) and consider text classification type's characteristics; thus, we aim to discover which factor is relevant to the text classification performance after eliminating or replacing it. To discover an answer to the aforementioned questions, we measured the text classification performance after eliminating or replacing particular factors of a given text. Supposing a particular factor is important for text classification performance, we presume a model trained with the training set where this factor is removed would acquire lower test classification performance. Therefore, we hypothesize a huge drop in test performance under the transformed training samples (removed factors) implies the importance of that factors.

We utilized three factors of text classification following the prior study (Garg & Ramakrishnan, 2020; Sun & Lu, 2020): **Probability-based Important Token**, **Attention-based Important Token**, **POS**, and **Syntax**. For **Probability-based Important Token**, we scored the importance of each token following the prior work of measuring token importance (Garg & Ramakrishnan, 2020). We measured the class probability change after removing every token and regarded a particular token as an important one if the removal causes the largest change among every token. For **Attention-based Important Token**, we observed each token's attention scores (Sun & Lu, 2020) on a given sentence and checked the token with the highest and lowest attention scores. For **POS**, we utilized NLTK toolkit, a conventional POS-tagger proposed in (Loper & Bird, 2002) to acquire POS information at each token. To select target POSs in our study, we extract various POS tags and count their ratio over total tokens in each dataset. The result is shown in Table 1. Among various POS tags, we selected nouns, verbs, and adjectives as target POSs as they frequently exist in every dataset. At the same time, the other POSs (i.e., prepositions, determiners) are absent or present with such low frequency, and we empirically expect these POS tags to influence less on the text classification; thus, we skipped these scarce POS tags in the analysis. Lastly, for **Syntax**, we randomly mixed the order of tokens in a sentence to provide a noise on the syntax of the given sentence, following the prior works of noising syntax (Wei & Zou, 2019). We removed and replaced these factors with training samples for each dataset, trained the classifier, and measured the test performance. The results are shown in Table 2.

**Composition of Important Tokens** Before we analyze important POS tags at each text classification type, we scrutinize the characteristics of the tokens that were considered important. We measured the ratio of different POS types existing in the important tokens followed by **Probability-based** approach. Note that we only described the composition of important tokens based on **Probability-based** approach as **Attention-based** approach showed very similar results to the aforementioned one. We only considered nouns, verbs, and adjectives as they were the majority of the POS tags. The

Table 1: The ratio of Top-12 POS tags exists in 9 benchmark datasets. Among various POS tags in the dataset, we utilize Top-3 POS tags (Nouns, Verbs, and Adjectives) in our study as they occur across every dataset with high frequency.

| | AGNews | Amazon | BANKING | DBPedia | IMDB | Ohsumed | r8 | StackOverFlow | Yelp | AVERAGE |
|---|---|---|---|---|---|---|---|---|---|---|
| **Nouns** | 0.3615 | 0.2139 | 0.3442 | 0.3587 | 0.2894 | 0.4519 | 0.4700 | 0.4743 | 0.4389 | **0.3781** |
| **Verbs** | 0.1189 | 0.1636 | 0.2824 | 0.1696 | 0.1529 | 0.2338 | 0.2350 | 0.2362 | 0.2275 | **0.2022** |
| **Adjectives** | 0.0770 | 0.0744 | 0.1159 | 0.0823 | 0.0784 | 0.1266 | 0.1319 | 0.1324 | 0.1250 | **0.1049** |
| **, (comma)** | 0.0320 | 0.0552 | 0.0964 | 0.0627 | 0.0434 | 0.0649 | 0.0644 | 0.0649 | 0.0639 | 0.0609 |
| **. (period)** | 0.0336 | 0.0333 | 0.0504 | 0.0523 | 0.0418 | 0.0625 | 0.0620 | 0.0620 | 0.0584 | 0.0507 |
| **Determiners** | 0.1044 | 0.0931 | 0.0133 | 0.0799 | 0.0854 | 0.0005 | 0.0013 | 0.0012 | 0.0121 | 0.0435 |
| **Preposition** | 0.0797 | 0.0997 | 0.0144 | 0.0624 | 0.0876 | 0 | 0.0001 | 0.0007 | 0.0119 | 0.0396 |
| **Cadinal Digit** | 0.0178 | 0.0597 | 0.0096 | 0.0125 | 0.0492 | 0.0014 | 0.0006 | 0.0002 | 0.0091 | 0.0178 |
| **Coordinating Conjunction** | 0.0226 | 0.0341 | 0.0031 | 0.0192 | 0.0297 | 0 | 0.0011 | 0.0002 | 0.0053 | 0.0128 |
| **: (colon)** | 0.0288 | 0.0052 | 0.0076 | 0.0053 | 0.0060 | 0.0089 | 0.0089 | 0.0091 | 0.0087 | 0.0098 |
| **to** | 0.0299 | 0.0122 | 0.0008 | 0.0338 | 0.0086 | 0.0005 | 0.0004 | 0.0001 | 0.0016 | 0.0098 |
| **Adverbs** | 0.0221 | 0.0216 | 0.0061 | 0.0090 | 0.0173 | 0 | 0 | 0.0004 | 0.0030 | 0.0088 |

Table 2: The classification performance after removing tokens with each respective condition. The values highlighted in bold indicate the greatest performance drop from the **Baseline** without any removal, also understandable as the most contributing factor in the task of text classification.

| | Topic Classification | | | | | | Sentiment Analysis | | |
|---|---|---|---|---|---|---|---|---|---|
| | StackOverFlow | BANKING | DBPedia | r8 | AGNews | Ohsumed | IMDB | Amazon Polarity | Yelp |
| **Baseline (No Removal)** | 0.8995 | 0.9276 | 0.9554 | 0.9410 | 0.9369 | 0.6338 | 0.9372 | 0.9522 | 0.9529 |
| **Probability-based** | 0.7405 | **0.6929** | **0.7906** | 0.7113 | **0.8301** | 0.4335 | 0.6448 | 0.8774 | 0.9507 |
| **Attention-based** | 0.7806 | 0.7023 | 0.8162 | 0.7345 | 0.8623 | **0.4330** | 0.6991 | **0.8338** | **0.8938** |
| **Nouns** | **0.2941** | 0.7268 | 0.8238 | **0.3761** | 0.9068 | 0.5282 | 0.9306 | 0.9472 | 0.9687 |
| **Verbs** | 0.8890 | 0.8331 | 0.9553 | 0.8165 | 0.9339 | 0.6145 | **0.3333** | 0.9462 | 0.9637 |
| **Adjectives** | 0.8763 | 0.8613 | 0.9517 | 0.8885 | 0.9344 | 0.5903 | 0.9218 | 0.9501 | 0.9699 |
| **Syntax** | 0.8990 | 0.9200 | 0.9359 | 0.7410 | 0.9280 | 0.5881 | 0.9350 | 0.9069 | 0.9469 |

Table 3: Ratio of the three POS tags in probability-based important tokens for each dataset. The values highlighted in bold indicate the POS tag that covers the greatest percentage. In sentiment analysis Verbs and Adjectives are more frequently considered important, except for the case in Yelp, where Nouns take the same ratio as Adjectives. We presume the reason as the high frequency of words such as *price, diner*, or restaurant names.

| | Topic Classification | | | | | | Sentiment Analysis | | |
|---|---|---|---|---|---|---|---|---|---|
| | StackOverFlow | BANKING | DBPedia | r8 | AGNews | ohsumed | IMDB | Amazon Polarity | Yelp |
| **Nouns** | **0.5940** | **0.4468** | **0.6374** | **0.6459** | **0.5369** | **0.5078** | 0.3352 | 0.3571 | **0.3743** |
| **Verbs** | 0.2334 | 0.4054 | 0.2417 | 0.1511 | 0.2963 | 0.2623 | 0.2681 | **0.3980** | 0.2513 |
| **Adjs** | 0.1725 | 0.1478 | 0.1209 | 0.2029 | 0.1668 | 0.2298 | **0.3967** | 0.2449 | **0.3743** |

results are summarized in Table 3. We could observe that while nouns are the most dominant POS tags for datasets under topic classification, verbs and adjectives become more critical for datasets for sentiment analysis.

**Analysis** Considering experiment results in Table 2 and the composition of important tokens at Table 3, we discovered that important POS information differs along with text classification types. For topic classification tasks, nouns created the most significant performance drop when removed, indicating that nouns play an essential role in the classification. On the other hand, verbs and adjectives contributed to a huge performance drop for sentiment analysis tasks. In the case of IMDB and Amazon Polarity datasets, tokens under the verb group created a huge drop when removed, and tokens under the adjective group made the largest drop when removed on the Yelp dataset. In a nutshell, we figured out that topic classification and sentiment analysis take different important POS types while they are conventionally grouped as a text classification task. Our study utilized this discovery as a concrete milestone in data augmentation techniques described in the following sections.

## 5 Do Augmenting Important POS Elevate Classification Performance?

**Setup** We hypothesize that words belonging to important POS tags contribute more to building the semantics of given text input. Also, replacing such words holds a higher risk of disrupting the label of the sentence. We, therefore, believe that replacing words that do not belong to important POS tags would yield a better classification performance escalation. For a detailed description of replacement-

based augmentation, we followed the method proposed in (Wei & Zou, 2019). Given a single input text, we created $n$ augmented samples by replacing $m$ tokens belonging to the particular POS tag (nouns, verbs, and adjectives) with synonyms from WordNet. We followed the same configuration (number of $n, m$) proposed in (Wei & Zou, 2019) as it is known to achieve the best performance in their experiment settings. Given training samples of every dataset, we trained the classifier by adding augmented training samples. For the target POS, we select important and non-important tokens for each dataset following the analyses in the prior section (which are denoted as *Important token Replacement* and *Non-important token Replacement*, respectively). Moreover, we utilize three POS types (Noun, Verb, and Adjective) as target POS information (which are denoted as *Noun Replacement*, *Verb Replacement*, and *Adjective Replacement*, respectively.) Except for augmenting training samples, we utilized the same experiment setting with the procedures in section 4. The experiment result is shown in Table 4.

Table 4: Classification performances based on augmentation of each condition, where the values highlighted in bold indicate the highest improvement compared to the baseline performance. Note that both **Important token replacement** and **Non-important token replacement** options are considered with **Probability-based** approach presented in section 4.

| | Topic Classification | | | | | | Sentiment Analysis | | |
|---|---|---|---|---|---|---|---|---|---|
| | StackOverFlow | BANKING | DBPedia | r8 | AGNews | ohsumed | IMDB | Amazon Polarity | Yelp |
| Baseline (No Augmentation) | 0.8995 | 0.9276 | 0.9554 | 0.9410 | 0.9369 | 0.6338 | 0.9372 | 0.9522 | 0.9529 |
| Important token Replacement | 0.8288 | 0.9229 | 0.9485 | **0.9516** | 0.9382 | 0.7128 | 0.9214 | 0.8478 | 0.9433 |
| Non-important token Replacement | 0.8988 | **0.9319** | 0.9560 | 0.9430 | **0.9390** | **0.7173** | 0.9331 | 0.9688 | 0.9614 |
| Noun Replacement | 0.8956 | 0.9192 | 0.9520 | 0.7645 | 0.9360 | 0.5698 | **0.9384** | 0.9529 | 0.9592 |
| Verb Replacement | 0.9003 | 0.9284 | **0.9584** | 0.9006 | 0.9388 | 0.6361 | **0.9384** | **0.9720** | **0.9644** |
| Adjective Replacement | **0.9023** | 0.9213 | 0.9577 | 0.8757 | 0.9324 | 0.6361 | 0.9382 | 0.9628 | 0.9622 |

**Analysis** We analyze that the replacement of tokens that do not belong to important POS tags does help in enhancing the classification performance for both classification tasks. The replacement of verbs instead of nouns produced the best results although the noun was the most important POS in a topic classification task. This is because the topic usually exists within the sentence in the form of a noun. An example of such a case would be as such: An original text from the AGNews dataset "Soaring crude *prices* plus worries about the *economy*" can be augmented as "Soaring crude *terms* plus worries about the *thriftiness*". The original text has the label "Economy", but the augmented sentence loses the factor to classify it under the "Economy" label. Furthermore, certain datasets possess domain-specific terms. For example, an example of a sentence from the StackOverFlow dataset is "Use *Oracle* 6 from ASP.NET application". After noun replacement, the sentence becomes "Use *prophet* 6 from ASP.NET application.". The original label for the text is "Oracle", and under the consensus that the term "Oracle" falls under a named entity that holds a technical meaning instead of the commonly known synonym of "prophet", the overall semantics of the sentence is completely altered.

However, we further discovered in sentiment analysis tasks that while verbs and adjectives are label determinants for the task, replacing such tokens does not reduce the classification performance but outputs the best results. We explain that the underlying reason for such behavior lies in the preservation of semantics before and after replacing a token. Unlike topic classification, the replacement of such words with their synonyms is less likely to alter the semantics of the word itself or the sentence as a whole. For example, the sentence "One of the other reviewers has *mentioned* that after *watching* just 1 Oz episode you'll be hooked" can be augmented as "One of the other reviewers has *remarked* that after *seeing* just 1 Oz episode, you'll be hooked", with the replaced word in italics. It can be understood that the sentiment information of the sentence is not affected as much as in the case of topic classification after replacing important POS tags, which are verbs and adjectives.

Consequentially, we discover that replacing verbs and adjectives (which are key factors in sentiment analysis tasks) enhances the classification performance in both topic classification and sentiment analysis at the same time. We analyze this result occurs because replacing verbs and adjective tokens is less likely to ruin the semantics of original text input when replaced with a synonym.

## 6 OUR APPROACH: PMIXUP

**Description** From prior analyses, we scrutinized that replacing verbs and adjectives with synonyms creates an effective data augmentation impact regardless of text classification tasks. We further

presume combining the feature space interpolation augmentation method with the aforementioned replacing verbs and adjectives would escalate classification performance rather than utilizing only either one of the methods. Therefore, we hereby present Part-of-speech replacement and Mixup (PMixUp), a novel data augmentation strategy that uses both synonym replacement and feature space interpolation. Please refer to Figure 1 for an overall architecture of our PMixUp. Our study is inspired by the works of (Wei & Zou, 2019) and (Chen et al., 2020b), but differs from them as we combined replacement-based and feature space interpolation methods for further escalation of test performance rather than using either of them. The proposed PMixUp merges two augmentation strategies: 1) replacing tokens belonging to verbs and adjectives with their synonyms in Word-Net, 2) TMix, which is a state-of-the-art feature space interpolation technique under circumstances using supervised learning with labeled data only (Chen et al., 2020b). In the following sections, we describe a series of empirical analyses to examine the effectiveness of PMixUp compared to replacement-based and feature space interpolation augmentation techniques.

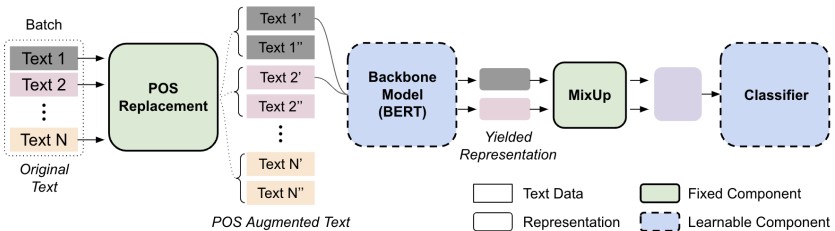

Figure 1: Overview of our PMixUp

**Experiment Setup** Following the setups in TMix study (Chen et al., 2020b), we measured the test classification performance under the different number of original training samples. We aim to examine whether the proposed PMixUp contributes to the escalation of test performance under the circumstance where a few original training samples exist. We set the number of original training samples per class as 10, 200, and 2500 if every class of the dataset includes more than 2500 training samples (AGNews, DBPedia, Amazon Polarity, Yelp, IMDB). For the datasets where each label includes less than 2500 samples, we heuristically set the number of training samples per class differently to describe a circumstance in which a few original training samples exist (StackOverFlow, BANKING, r8, Ohsumed).

Throughout the experiment, we utilized three data augmentation techniques: **Synonym Replacement**, **Feature Space Interpolation**, and **PMixUp**. For synonym replacement, we measured the test classification performance under the replacement-based augmentation method proposed in section 5, which replaces tokens belonging to nouns, verbs, and adjectives with synonyms in Word-Net. For the feature space interpolation method (which is currently state-of-the-art), we followed the same configuration in the original publication of TMix. We interpolated the latent representation at one of the 7th, 9th, and 12th layers of the model, and we denoted both the reproduced performance and the reported performance in the original publications. Lastly, for PMixUp, the augmented data from synonym replacement is passed to the TMix module to apply feature space interpolation into the given sample. Furthermore, in order to obtain a detailed understanding of the effect of POS-based augmentation, we compare the results with those of augmentation through random token replacement without utilzing any POS information. Note that we skipped experiments on previously-proposed replacement-based and feature space interpolation methods as they were inferior to TMix in the public benchmark settings. The experiment results following above setups are summarized in Table 5 and Table 6.

**Analysis** From the experiment results shown in Table 5 and Table 6, we figured out the proposed PMixUp mostly outperforms the classification performance rather than the case when only either of replacement-based and feature space interpolation method is used. Upon this supremacy of PMixUp, we figure out several takeaways. First and foremost, the effectiveness of PMixUp follows the trend analyzed in section 5. The replacement-based augmentation was most effective when we transformed tokens of verbs or adjectives regardless of text classification types. Accordingly, PMixUp

also performed best when we augmented tokens under verbs and adjectives before interpolating feature space with TMix.

Furthermore, PMixUp is especially effective when extremely few training samples exist per class. We presume this result implies that PMixUp creates augmented training samples with the lowest interference on the core semantics of given original training samples. Furthermore, we interpret the benefit of data augmentation (PMixUp as well as the other methods) as weakened when many training samples exist per class. A large number of training samples would be enough to learn the core semantics of each intent; thus, the augmentation would become less effective in creating benefits for the model. Accordingly, we discovered several cases where the solely-applied replacement-based augmentation method performs better than PMixUp. We evaluate the performance gap between the PMixUp and the best augmentation method as small (less than 1 or 2 percent). Nevertheless, we resulted in PMixUp becoming a state-of-the-art data augmentation method to escalate the text classification performance in general cases in our experiment settings. Thus, it would benefit machine learning practitioners and the NLP community to tackle the challenge of lack of training samples when they train their text classifiers.

Table 5: Performance comparison on topic classification tasks. Note that **Important Token** option is considered with **Probability-based** approach presented in section 4.

| | | AGNews | | | DBPedia | | | StackOverFlow | | | BANKING | | r8 | | Ohsumed |
|---|---|---|---|---|---|---|---|---|---|---|---|---|---|---|---|
| | | 10 | 200 | 2500 | 10 | 200 | 2500 | 10 | 200 | 600 | 10 | 30 | 10 | 32 | 8 |
| **Baseline (No Augmentation)** | | 0.7050 | 0.8822 | 0.9080 | 0.9500 | 0.9861 | 0.9900 | 0.2668 | 0.8755 | 0.9012 | 0.8918 | 0.9173 | 0.9222 | 0.9400 | 0.6812 |
| **Synonym Replacement** | **Noun** | 0.6607 | 0.7864 | 0.9080 | 0.9500 | 0.9854 | 0.9802 | 0.5610 | 0.8795 | 0.9020 | 0.8455 | 0.9187 | 0.9511 | 0.7645 | 0.6977 |
| | **Verb** | 0.7129 | 0.7905 | 0.9080 | 0.9749 | 0.9860 | 0.9900 | 0.6890 | **0.8808** | 0.9021 | 0.8324 | 0.9201 | 0.9628 | 0.9006 | 0.6968 |
| | **Adjective** | 0.6846 | 0.7892 | 0.9080 | 0.9746 | 0.9854 | 0.9900 | 0.5292 | 0.8778 | 0.8910 | 0.8945 | 0.9115 | 0.9621 | 0.8757 | **0.7112** |
| | **Important Tokens** | 0.7023 | 0.7892 | 0.9100 | 0.9443 | 0.9820 | 0.9823 | 0.5418 | 0.8651 | 0.8878 | 0.8077 | 0.8349 | 0.9408 | 0.8401 | 0.6680 |
| | **Random Tokens** | 0.7060 | 0.8818 | 0.9121 | 0.9018 | 0.9863 | 0.9909 | 0.4987 | 0.8762 | 0.8968 | 0.7932 | 0.8818 | 0.9790 | 0.9622 | 0.6967 |
| **TMix (reported)** | | 0.7410 | 0.8810 | 0.9100 | 0.9680 | 0.9870 | 0.9900 | - | - | - | - | - | - | - | - |
| **TMix (reproduced)** | | 0.7647 | 0.8822 | 0.9100 | 0.9651 | 0.9884 | 0.9900 | 0.0663 | 0.8762 | 0.9010 | 0.7042 | 0.8991 | 0.9888 | 0.9516 | 0.5611 |
| **PMixUp (OURS)** | **Noun** | 0.7796 | 0.7913 | 0.9179 | 0.9622 | 0.9867 | **0.9913** | 0.7319 | 0.8775 | 0.9010 | 0.8919 | 0.9012 | 0.9652 | 0.9430 | 0.6712 |
| | **Verb** | 0.8014 | **0.8828** | **0.9204** | **0.9763** | **0.9884** | **0.9913** | **0.7841** | 0.8797 | **0.9022** | **0.9001** | **0.9212** | **0.9889** | **0.9617** | 0.7109 |
| | **Adjective** | **0.8064** | 0.8817 | 0.9188 | 0.9685 | 0.9860 | **0.9913** | 0.7791 | 0.8738 | 0.9014 | 0.8911 | 0.9117 | 0.9525 | 0.9420 | 0.7088 |
| | **Important Tokens** | 0.8023 | 0.8822 | 0.9100 | 0.9697 | 0.9861 | 0.9900 | 0.7546 | 0.8512 | 0.8978 | 0.8504 | 0.8645 | 0.9419 | 0.8911 | 0.6708 |
| | **Random Tokens** | 0.7762 | 0.8818 | 0.9118 | 0.9371 | 0.9868 | 0.9888 | 0.6463 | 0.8763 | 0.8975 | 0.6117 | 0.8763 | 0.9733 | 0.9632 | 0.6879 |

Table 6: Performance comparison on sentiment analysis tasks. The highlighted values indicate the highest scores achieved in a particular setting. Note that **Important Token** option is considered with **Probability-based** approach presented in section 4.

| | | Amazon Polarity | | | Yelp | | | IMDB | | |
|---|---|---|---|---|---|---|---|---|---|---|
| | | 10 | 200 | 2500 | 10 | 200 | 2500 | 10 | 200 | 2500 |
| **Baseline (No Augmentation)** | | 0.6148 | 0.869 | 0.8887 | 0.7128 | 0.8928 | 0.9122 | 0.6430 | 0.8796 | 0.9132 |
| **Synonym Replacement** | **Noun** | 0.6088 | 0.8796 | 0.9078 | 0.7078 | 0.8796 | 0.9112 | 0.6498 | 0.8836 | 0.9222 |
| | **Verb** | 0.6122 | 0.8836 | 0.9120 | 0.7388 | 0.8922 | **0.9239** | 0.6328 | 0.8854 | 0.9198 |
| | **Adjective** | 0.7173 | 0.8854 | 0.9198 | 0.7412 | 0.8854 | 0.9121 | 0.6328 | 0.8860 | **0.9202** |
| | **Important Tokens** | 0.6022 | 0.8860 | **0.9202** | 0.7128 | 0.8860 | 0.9009 | 0.6440 | 0.7960 | 0.9000 |
| | **Random Tokens** | 0.6276 | 0.8760 | `0.9017` | 0.7284 | 0.8845 | 0.9138 | 0.6262 | 0.8829 | 0.9220 |
| **TMix (reported)** | | - | - | - | - | - | - | 0.6930 | 0.8740 | 0.9030 |
| **TMix (reproduced)** | | 0.5142 | 0.7960 | 0.9000 | 0.7821 | 0.9411 | 0.9132 | 0.6986 | 0.8712 | 0.9076 |
| **PMixUp (OURS)** | **Noun** | 0.6768 | 0.8740 | 0.9030 | 0.7901 | 0.9511 | 0.9222 | 0.6340 | 0.8746 | 0.9110 |
| | **Verb** | **0.7221** | 0.8712 | 0.9076 | **0.8109** | **0.9619** | 0.9114 | **0.6988** | **0.8886** | 0.9108 |
| | **Adjective** | 0.6809 | **0.882** | 0.9110 | 0.7589 | 0.9617 | 0.9114 | 0.6854 | 0.8840 | 0.9116 |
| | **Important Tokens** | 0.7138 | 0.8655 | 0.9108 | 0.7645 | 0.9408 | 0.9089 | 0.6455 | 0.8230 | 0.8978 |
| | **Random Tokens** | 0.5649 | 0.8778 | 0.8999 | 0.6625 | 0.9180 | 0.9209 | 0.6450 | 0.8780 | 0.9079 |

# 7 WHAT DRIVES PMIXUP'S EFFECTIVENESS?

**Setup** Upon discovering the effectiveness of PMixUp from above sections, we further investigate the underlying reason for its superior performance. We hypothesize that a better learning paradigm would fully utilize the capacity of the model, which means that models achieving better inference performance contain more fruitful information within its layers. In order to justify our hypothesis, we utilize Centered Kernel Alignment (CKA) (Kornblith et al., 2019), a method which returns the similarity between two models in a scale between 0 and 1. We compare the representation similarities at different layers when the model is trained with different augmentation strategies. A higher similarity among layers within the same network would mean a shared knowledge embraced between the layers, further implying that the model has a low knowledge capacity and a lack of ability to describe more abundant aspects of a given text data. Note that among various methods to quantify

representation similarities, such as CCA (Hardoon et al., 2004) or SVCCA (Raghu et al., 2017), we utilize CKA which presented state-of-the-art performances in their domain's benchmark settings (Kornblith et al., 2019). For implementation details, we measured the layer-wise similarity within a model by extracting the feature vectors obtained after normalization of each fine-tuned BERT layers. We compared the results from a baseline model without any augmentation, TMix, and our proposed PMixUp. The test dataset was used for the CKA process, and the results are shown in Figure 2

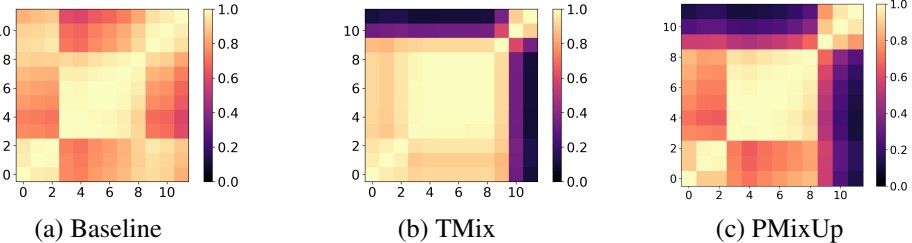

(a) Baseline        (b) TMix        (c) PMixUp

Figure 2: Visualized representation similarities yielded by CKA on BANKING dataset.

**Analysis** From the experiment results, we could confirm our initial hypothesis and observe that a model trained with the proposed PMixUp has the highest knowledge capacity among different augmentation strategies. While the other two setups showed higher similarities among layers, PMixUp resulted in a lower representation similarity. The difference was especially noticeable at higher levels of the model, which can connect to the prior works of (Rogers et al., 2020) and (Ethayarajh, 2019); while higher layers not only contain task-specific information but also high-level contextual understandings of texts, a clearly low similarity in such layers imply that a model trained with PMixUp is more capable of understanding the contextual information of a given text. We can hence conclude that a model trained with the PMixUp approach is able to illustrate given data in a more fruitful manner, and therefore leading to a high classification performance.

## 8    DISCUSSIONS AND CONCLUSION

We present a series of analyses to discover key factors of text classification and effective data augmentation methods. First, we figure out that important POS information differs in topic classification and sentiment analysis; nouns are critical determinants in topic classification, whereas verbs or adjectives are essential in sentiment analysis. Furthermore, we extended these observations by questioning whether replacing these important POS would escalate classification performance when it gets augmented. Throughout experimental analyses, we discovered that replacing verbs and adjectives was commonly effective in both topic classification and sentiment analysis tasks. Upon these findings, we further introduce PMixUp, a novel data augmentation strategy that utilizes both synonym replacement and feature space interpolation. We prove that our method not only matches but outperforms previously dominant methods through experiments on nine benchmark datasets, achieving state-of-the-art results, especially in a case where few training samples exist per class. While our study proposes concrete data augmentation practices, several improvement avenues still exist. While the PMixUp replaces particular POS tokens with synonyms in WordNet, future works might utilize other methods of finding synonyms (i.e., Thesaurus (Roget's 21st Century Thesaurus, 2013), mytheas (LibreOffice) (Zhang et al., 2015)). Future works might improve our PMixUp with the other feature space interpolation techniques such as Local Additivity based Data Augmentation (LADA) (Chen et al., 2020a). Furthermore, our PMixUp could be examined under harsher circumstances, such as class imbalance, to check its robustness. Lastly, we would examine the effectiveness of PMixUp in other NLP tasks such as Question Answering (Rogers et al., 2021), Named Entity Recognition (Yadav & Bethard, 2019), or translation (Yang et al., 2020a). We expect our study serves as an effective guideline for data augmentation, and recommend applying our works on various text classification tasks.

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
