# OpenReview forum: "PMixUp: Simultaneous Utilization of Part-of-Speech Replacement and Feature Space Interpolation for Text Data Augmentation"
_ICLR.cc/2023/Conference — Submitted to ICLR 2023_

### Official Review · Reviewer_mExN · 2022-10-24

**Confidence:** 4
**Correctness:** 3
**Technical Novelty And Significance:** 3
**Empirical Novelty And Significance:** 3
**Recommendation:** 6

**Clarity, Quality, Novelty And Reproducibility:**

This paper is relatively clear and the content is original. It might be hard to replicate the experiments since the details are fuzzy.

**Strength And Weaknesses:**

Strengths:

* This paper combines the replacement-based method and the feature space interpolation method together to increase the performance.
* This paper answers a lot of why questions instead of how questions.

Weaknesses:

* I don't think our field put topic classification and sentiment analysis into the same category.

**Summary Of The Paper:**

This paper proposes a new data augmentation method, PMixUp. Specifically:

* This paper performs a relatively comprehensive study of the importance of POS tags in topic classification and sentiment analysis.
* This paper proposes a new data augmentation method, PMixUp, that utilizes both replacement-based and feature space interpolation methods.
* The authors tests their new method on nine public benchmark settings.



**Summary Of The Review:**

I am leaning towards acceptance but would like to see some revisions to the paper.

---

> ### Author Response · Authors · 2022-11-12
> **Response to Reviewer mExN**
>
> Thank you for your detailed review and comments on our work. We are glad you find our paper's clear and original contributions to the research community. We would like to clarify your concerns and comments below. We also updated our manuscript with your concerns regarding the difficulty of replication by adding a figure of our framework, as well as clear details of our approach. We hope you consider raising your score if you find our response satisfactory.
>
> **Q. Don’t think our field put topic classification and sentiment analysis into same category.**
>
> We appreciate your comments regarding this concern. However, when we analyze many previously-proposed data augmentation studies ([1], [2], [3], [4], [5]), we discovered that most prior works regard the two tasks on a same basis. We conducted experiments on whether it is truly reasonable to consider the two tasks equally, and discovered that there exist differences between the two tasks.
>
> [1] Limsopatham, Nut. "Effectively leveraging bert for legal document classification." *Proceedings of the Natural Legal Language Processing Workshop 2021*
> . 2021.
>
> [2] Feng, Steven Y., et al. "Genaug: Data augmentation for finetuning text generators." *arXiv preprint arXiv:2010.01794*
>  (2020).
>
> [3] Wei, Jason, and Kai Zou. "Eda: Easy data augmentation techniques for boosting performance on text classification tasks." *arXiv preprint arXiv:1901.11196*
>  (2019).
>
> [4] Chen, Jiaao, Zichao Yang, and Diyi Yang. "Mixtext: Linguistically-informed interpolation of hidden space for semi-supervised text classification." *arXiv preprint arXiv:2004.12239*
>  (2020).
>
> [5] Jin, Di, et al. "Is bert really robust? a strong baseline for natural language attack on text classification and entailment." *Proceedings of the AAAI conference on artificial intelligence*
> . Vol. 34. No. 05. 2020.
>
>
> **Q. It might be hard to replicate the experiments since the details are fuzzy**
>
> Thank you for your comment. In the revised manuscript, we added a detailed figure of the overall architecture in Figure 1 for a more accurate understanding of our PMixUp.

---

### Official Review · Reviewer_iN19 · 2022-10-24

**Confidence:** 3
**Correctness:** 4
**Technical Novelty And Significance:** 4
**Empirical Novelty And Significance:** 4
**Recommendation:** 8

**Clarity, Quality, Novelty And Reproducibility:**

The proposed approach is nuanced, but the algorithm for replicating this is unclear. If possible, perhaps the author can briefly discuss how PMxiUp can be reproduced. Maybe very difficult to implement compared to Emix, a data augmentation method that uses interpolations of word embeddings and hidden layer representations to construct virtual examples (https://aclanthology.org/2020.coling-main.611/).

**Details Of Ethics Concerns:**

No comments

**Strength And Weaknesses:**

An exciting new text method is presented in text data augmentation, focusing on using different POS correlates to the performance of topic classification and sentiment analysis; nouns are critical determinants in topic classification, whereas verbs or adjectives are essential in sentiment analysis. The results show that the proposed model using CKA during training performs well when comparing the representation similarities at different layers when the model is trained with different augmentation strategies scaling between 0 and 1.

Some assertions are made that are not entirely supported by the results obtained. Maybe this is how I felt because the results did not thoroughly analyze how the proposed method would perform with the class imbalance dataset and whether it would be inadequate for a robust model.

Many why questions instead of the how ones are addressed in this article.

**Summary Of The Paper:**

This paper describes PMxiUp, a new way to add to text data augmentation by replacing parts of speech (POS) and interpolating features. The proposed method swaps out tokens that belong to a particular POS tag and uses feature space interpolation in order. The author discovered through experimentation that nouns are essential in determining how to label a topic. At the same time, verbs and adjectives are paramount in figuring out how someone feels about something. In the reported experiments on nine public benchmark datasets to test PMixUp's best performance on classification performance rather than previous works, it becomes a new state-of-the-art in specific settings.

**Summary Of The Review:**

Good: This paper is of interest to the *ICLR audience and could be published, but might not be appropriate for a top-tier publication.

---

> ### Author Response · Authors · 2022-11-12
> **Response to Reviewer iN19**
>
> Thank you for your thorough review and thoughtful comments. We are glad you find our PMixUp as an effective augmentation technique. We would like to clarify your concerns below. We updated our manuscript according to the suggestions and comments. We hope you consider raising your score if you find our response satisfactory.
>
> **Q. The results did not thoroughly analyze how the proposed method would perform with class imbalance dataset and whether it would be inadequate for a robust model.**
>
> Thank you for raising this concern. We state that our experiments followed public benchmark settings, which do not consider imbalanced datasets. We conducted experiments following the setups of prior works on text augmentations ([1], [2]). These works conducted experiments on public benchmark settings, without considering settings with class imbalance. Still, we agree that further experiments under the class imbalance can provide more fruitful takeaways to the research community. We strongly believe it to be a significant improvement avenue, and we will address the necessity of these experiments in the Discussions section for future works. We sincerely appreciate the reviewer for commenting on this point.
>
> [1] Chen, Jiaao, Zichao Yang, and Diyi Yang. "Mixtext: Linguistically-informed interpolation of hidden space for semi-supervised text classification." *arXiv preprint arXiv:2004.12239* (2020).APA
>
> [2] Wei, Jason, and Kai Zou. "Eda: Easy data augmentation techniques for boosting performance on text classification tasks." *arXiv preprint arXiv:1901.11196*
>  (2019).
>
> **Q. Algorithm for replicating is unclear. Briefly discuss how PMixUp can be reproduced.**
>
> Thank you for your suggestion. In the revised manuscript, we added our PMixUp's overall architecture in Figure 1. Please refer to the attached revised manuscript for the improved version. We also added implementation details and codes in the supplementary materials for reproducibility.
>
> **Q. Difficult to implement compared to EMix.**
>
> We appreciate your reference for comparison, but would like to highlight the efficiency brought by adding one simple module beforehand. The suggested EMix model has the same overall architecture as TMix, which interpolates feature space within the layers of the Language Model. On that line, our proposed PMixUp requires an additional step to identify and replace POS tokens prior to the feature interpolation step, which could add difficulty to the implementation compared to TMix or EMix. However, we believe that this additional step and effort can be relatively easy to implement with the help of pre-implemented libraries such as NLTK. Please refer to Figure 1 in the revised manuscript for a more visualized understanding of the PMixUp.

---

### Official Review · Reviewer_8rka · 2022-10-24

**Confidence:** 4
**Correctness:** 3
**Technical Novelty And Significance:** 2
**Empirical Novelty And Significance:** 2
**Recommendation:** 3

**Clarity, Quality, Novelty And Reproducibility:**

Clarity: Mostly clear.  Additional contextualization for the analyses would be helpful.  The claims made in the abstract and introduction are less consistently supported by the data than would be expected.

Quality: Reasonably high

Novelty: Modest.  Both TMix and replacement augmentation have been previously developed. Their combination is being addressed in this work, with additional analysis for the tasks the paper focuses on.

Reproducibility: High



**Strength And Weaknesses:**

Strengths
* This is a reasonable combination of existing techniques.
* Decent analyses of different POS influence on performance of different tasks on a variety of corpora.

Weaknesses
* The identification of which POS is relevant for which task remains very manually derived. It would be interesting to see the conditioning on POS as part of an objective function that could be learned as part of the training operation. This would eliminate the need for a user to perform a POS base analysis in order to bring this framework to a new task.
* The reported claims of NOUNS for topic classification and VERBS and ADJECTIVES for sentiment is not as clearly supported by the data as the abstract would suggest.  While the role of nouns in topic classification is reasonably well supported, Table 3 suggests a more nuanced situation in the Sentiment Analysis corpora with less consistency across corpora.  Similarly Table 4 suggests that there are corpora differences again in the value of replacing different POS classes.  While the syntactic class as a space for augmentation.
* The reported differences in Tables 5 and 6 are small.  Some significance testing would go a long way to understanding the relevance of these findings.  Specifically, it's not clear that the value of PMixUp compared to TMix is substantial enough to warrant the additional training and risk of poor replacement generation.
* The findings from Tables 1-3 seem to be inconsistent with those in Table 5.  The replacement based augmentation benefits from NOUNS for topic classification somewhat strongly.  However, in Table 5, this suggests that VERB replacement is better when replacement augmentation is combined with TMIX.  This relationship is surprising and suggests that initial analyses based on, say replacement, may not be useful to designing a system.  The two approaches aren't entirely complementary

**Summary Of The Paper:**

This manuscript combines two reasonable augmentation strategies to improve topic classification and sentiment analysis.  The first is replacement based augmentation and TMix style feature space augmentation.

The findings suggest that using Part of speech (POS) to drive replacement augmentation matters, and its impact is different based on the task.  Second, these two strategies combine effectively.

**Summary Of The Review:**

The approach of combining TMix and replacement augmentation is sensible.  The particular claims about the relationship between POS classes and sentiment analysis are not as clean as would be expected from the presentation of the work.  It's not clear how this would be extended to some other NLP/NLU task without a similar precursor POS analysis.  Additionally the value of POS sensitive replacement is not obviously necessary for either task; while there is some improvement, it is typically quite modest and not entirely consistent across tasls .

Table 6 includes a typo 9017 should be 0.9017.
Table 5 why is StackOverflow 10, TMix (reproduced) only 0.0663 while even the no augmentation baseline is 0.2668?

---

> ### Author Response · Authors · 2022-11-12
> **Response to Reviewer 8rka**
>
> We appreciate your detailed comments and hereby carefully address each of your concerns as follows. We also updated our manuscript according to the suggestions. We hope you consider raising your score if you find our response satisfactory.
>
> **Q. Manually-derived relevant POS identification**
>
> However, we emphasize that the objective of the pointed analysis was to empirically prove the difference each POS holds in different text classification tasks. Such analysis has not been conducted in prior works, and we had to manually examine the importance of each POS in order to statistically exhibit the results (as a preliminary analysis). Still, we strongly agree that the POS information can be parameterized to be learned as part of the training operation, and we believe it can serve as an important improvement avenue in future works.
>
> **Q. Inconsistency in POS importance for Sentiment Analysis across corpora**
>
> We presume the underlying reason is in the task's nature. We expect classifying sentiment is a more sophisticated, complex task when we compare it with topic classification. For example, Verbs such as "like" can be as important as Adjectives such as "good" in determining the sentiment semantics of a given sentence. The Verb can be a solely important one; the Adjective also can be a solely important one, or these POS information are simultaneously significant for classifying sentiment. Conversely, we presume Nouns (i.e., haskell, oracle in STACKOVERFLOW) can be the sole key factor for classifying the topic; thus, it causes a consistent result in topic classification (as the reviewer mentioned).
>
> **Q. Clarifications on Table 3 and 4**
>
> We emphasize that the POS importance shown in Table 3 does not guarantee escalation of classification performance when it gets replaced, as shown in Table 4.
>
> In Table 3, we aim to illustrate important POS information at different text classification tasks. We observe Nouns were important in topic classification, while Verbs or Adjectives were significant to sentiment analysis tasks. However, we wonder whether augmenting these important POS tokens directly relates to the escalation of classification performance; thus, we conduct the follow-up analyses in Section 5 with Table 4.
>
> When we analyzed it, interestingly, we figured out that replacing the aforementioned important POS tokens was not related to the classification performance escalation. Instead, replacing Verbs or Adjectives was commonly effective at general classification tasks. Accordingly, this also becomes a motivation for our PMixUp.
>
> **Q. Inquiry on Significance testing**
>
> We agree that statistical significance is important in justifying our work’s relevance. Instead of statistical testing (i.e., p-value), which highly depends on heuristically-established null hypothesis, we report the average of three runs for the statistical significance. We expect this averaging three trials is sufficient enough as most prior works ([1]) followed this scheme for justifying their statistical significance.
>
> [1] Chen, Jiaao, et al. "Mixtext: Linguistically-informed interpolation of hidden space for semi-supervised text classification."
>
> **Q. PMixUp's performance is substantial enough?**
>
> We respectfully disagree, and emphasize the performance enhancement for all cases except Yelp with 2500 samples per class. We argue that this proves the consistent efficiency of our approach over TMix across different datasets and settings. Especially we also highlight PMixUp's effectiveness under a few samples per class compared to TMix.
>
> **Q. Inconsistent results from Tables 1-3 and Table 5**
>
> We would like to emphasize one of our findings that important POS does not necessarily serve as an effective POS to augment by replacement. Tables 1-3 are results of discovering the important POS itself, instead of replacing them. The results serve as empirical preliminary analysis to discover the important POS respective to each task. On the contrary, the results in Tables 5 (and 6) serve to support our claim that instead of replacing important POS, replacing verbs or adjectives that least affect the semantics of a given sentence is the key to efficient text augmentation.
>
> **Q. Typo in Table 6**
>
> We have corrected our typo in Table 6 from 9017 to 0.9017.
>
> **Q. Low performance of TMix for STACKOVERFLOW 10**
>
> We hereby say that TMix is not an augmentation strategy with low samples per class, especially under STACKOVERFLOW. We had the same doubt over the results of TMix under STACKOVERFLOW 10 settings, even with repeated experiments. We believe that the reason lies in the words or tokens that has domain-specific meanings such as haskell, oracle, which hinders the model from forming an effective representation under few samples settings. Additionally, interpolating such disqualified feature space would have further affected the resulting performance. Please refer to the supplemented code implementations for a more intuitive observation on our results.

---

### Official Review · Reviewer_Rbzx · 2022-12-03

**Confidence:** 3
**Correctness:** 3
**Technical Novelty And Significance:** 2
**Empirical Novelty And Significance:** 2
**Recommendation:** 5

**Clarity, Quality, Novelty And Reproducibility:**

Clarity and Quality:
The paper is clear and the content is easy to read.

Novelty:
The novelty of the proposed method is relatively limited, especially in terms of theoretical contribution.

Reproducibility:
Due to the lack of details of the experimental setup, it might be difficult to reproduce the experiment.

**Strength And Weaknesses:**

Strengths:
* The paper reads well and it is in general easy to follow.
* The idea is neat and easy to implement the method of combining the two augmentations is clear.

Weaknesses:
* The novelty of the proposed method is relatively limited, especially in terms of theoretical contribution.
* Determining which types of POS are relevant to model performance on push tasks is still somewhat unclear. It would be great to further explore the error analysis based on the tokens and sentences.
* The impact of model performance by augmenting Nouns, Verbs and Adjectives for those NLP tasks are limited from the results shown in Table 3 and Table 4.

**Summary Of The Paper:**

This paper proposes PMixUp that combines two augmentation methods (replacement-based augmentation and feature space augmentation) for the topic classification and sentiment analysis. They emphasis the importance of POS tags for those NLP tasks based on their study, and apply the feature space interpolation to swap out tokens corresponding to those POS tags. The results indicate that nouns are matter to some extent when selecting to label the topics.

**Summary Of The Review:**

The method of combining two augmentation of techniques is reasonable. The results show the types of POS can impact for the model performance in some circumstances. However, it is not clear to demonstrate the settings for all the experiments.

---

> ### Author Response · Authors · 2022-12-07
> **Response to Reviewer Rbzx**
>
> We appreciate your detailed comments on our work, and are glad you find our work easy to implement and clear. We hereby address your concerns as follows:
>
> **Q. The novelty of the proposed method is relatively limited, especially in terms of theoretical contribution.**
>
> We understand that the reviewer might doubt the novelty of our proposed method. However, we would like to emphasize our work being the first to  understand the POS essential to each topic classification and sentiment analysis tasks.
> Secondly, we discovered the common trend in terms of  augmentation through POS replacement regardless of the task type, and analyzed the underlying reasons for the pattern. Lastly, we combined the above observations with feature space interpolation to result in a universal text augmentation strategy outperforming existing works.
>
> We therefore believe our contributions can benefit the NLP community in gaining a more solid understanding of different text classification tasks.
>
>
> **Q. Determining which types of POS are relevant to model performance on push tasks is still somewhat unclear. It would be great to further explore the error analysis based on the tokens and sentences.**
>
> We respectfully disagree; the important POS for each task are summarized in Table 2 and the Analysis of Section 4. Removing nouns caused the greatest performance drop for Topic Classification tasks, and removing verbs caused the greatest performance drop for Sentiment Analysis tasks. Please note that this is an analysis of the important POS itself, not the POS to augment.
>
> Still, we agree that a further error analysis based on tokens and sentences can provide more clear insights of our analysis.
>
>
>
> **Q. The impact of model performance by augmenting Nouns, Verbs and Adjectives for those NLP tasks are limited from the results shown in Table 3 and Table 4.**
>
> We point out that Table 3 does not contain results of any augmentation, but shows the ratio of different POS words in probability-based important tokens. Table 4 contains the performance results after augmenting each conditions. When observing only the POS information (last 3 rows), a more clear common trend is visible that augmenting verbs generally gives the best results.
>
> **Q. Due to the lack of details of the experimental setup, it might be difficult to reproduce the experiment.**
>
> The experimental details of the experiments are listed in the paper, and the code implementation is provided for convenience. Please refer to the supplementary code implementations for further details of the experiments.
>
>
> **Additional comment:** Thank you for your thorough comments, and hope you might consider raising your scores should you find our response satisfactory.

---

### Author Response · Authors · 2022-11-12
**Response to All Reviewers**

We sincerely appreciate every reviewer for valuable comments on our work. We are glad that the reviewers generally find our paper well-written with meaningful contributions to the research community.

We addressed your comments regarding our work in your review sections. Please have a look, and let us know if there are any points of clarification or additional concerns about our work or responses. We will follow up as soon as possible for your decision. We hope you raise your score if our responses are satisfactory.

We sincerely hope that this work will benefit the communities of machine learning and NLP. We have made some changes to address the comments from the reviewers. We would really appreciate it if the reviewers checked our revised manuscript.

---

### Decision · Program_Chairs · 2023-01-20

**Decision:**

Reject

**Justification For Why Not Higher Score:**

Reviewers experience was taken into account in arriving at the final Reject.  The one Accept had less confidence and substantially less publication experience than the one Reject.  On close inspection of the reviews, 8rka was found to have good points that weren't well rebutted.

**Justification For Why Not Lower Score:**

N/A

**Metareview: Summary, Strengths And Weaknesses:**

The paper combines three augmentation methods (replacement-based augmentation which is enabled using POS tags, feature space augmentation and a Mixup style) to support training for text classification tasks.  They did empirical analysis to understand how important tokens of different POS tags were for classification.  They also tested their approach on a wide variety of data sets including sentiment analysis.  This is good work but contested by reviewers.

The novelty is relatively limited.  The relative role of nouns, verbs and adjectives has almost certainly been studied before in text classification, but the results (Table 4) are mixed other than the favorability of verbs. The use of POS with augmentation is the key novelty.  The empirical evaluation of Tables 1&2 was inconsistent with the eventual results, Table 5&6 so less useful.

In text classification, a common standard for significance testing is average of 5 or 10.  10 is adequate to do hypothesis testing per data set.  With less you need one hypothesis test for the full experiment.   [1] is a reference for augmentation but not a classic text classification work.  Also, the model is developed (i.e., the POS sets to use), i.e., as shown in Table 4, using the same data sets used in the final evaluation, i.e., Tables 5 & 6.   The final Tables 5&6 should only have Verb, the best performing, it seems poor to present 5 different methods and then report the best performing.

There was a substantial difference between reviewers.  In interpreting this, reviewers confidence, experience and publications were taken into account in arriving at the final Reject.

**Summary Of Ac-Reviewer Meeting:**

N/A